# The Role of Wilms’ Tumor Gene (WT1) Expression as a Marker of Minimal Residual Disease in Acute Myeloid Leukemia

**DOI:** 10.3390/jcm11123306

**Published:** 2022-06-09

**Authors:** Davide Lazzarotto, Anna Candoni

**Affiliations:** Division of Hematology and Stem Cell Transplantation, ASUFC, University of Udine, 33100 Udine, Italy

**Keywords:** WT1, minimal residual disease, acute myeloid leukemia

## Abstract

The Minimal Residual Disease(MRD) monitoring in acute myeloid leukemia (AML) is crucial to guide treatment after morphologic complete remission, to define the need for consolidation with allogeneic stem cell transplantation (Allo-SCT), and to detect impending relapse allowing early intervention. However, more than 50% of patients with AML lack a specific or measurable molecular marker to monitor MRD. We reviewed the key studies on WT1 overexpression as a marker of MRD in AML patients undergoing an intensive chemotherapy program, including Allo-SCT. In addition, we provided some practical considerations on how to properly use WT1 expression as an MRD marker, considering its strengths and weaknesses. In order to achieve the best sensitivity and specificity, it is recommended to refer to the standardized method of European LeukemiaNet and its defined threshold (250 WT1 copies/10^4^ Abelson (ABL) on Bone Marrow-BM and 50 WT1 copies/10^4^ ABL on Peripheral Blood-PB), which has been validated in a large and multicenter cohort of patients and normal controls.

## 1. Introduction

The detection of minimal residual disease (MRD) in acute myeloid leukemia (AML) has an important role in risk stratification and treatment planning. However, more than 50% of patients with AML lack a specific or measurable molecular marker to monitor MRD. To date, only four leukemia-specific genes or fusion transcripts have been extensively studied and validated for this purpose, namely mutated NPM1, RUNX1-RUNX1T1, CBFB-MYH11, and PML-RARA [1,2].

Because the Wilms tumor gene (WT1) is overexpressed in more than 80% of AML, it has been studied as a potential marker of MRD, but its role in this context is still debated [1,2,3,4]. Furthermore, WT1 appears to be involved in the pathogenesis of at least some subgroups of AML, but the exact mechanisms are not yet fully understood as the prognostic significance of its overexpression in AML is not clear [5].

In this review, we summarize the key findings on the WT1 gene, with a particular focus on its role as a marker of MRD.

## 2. The WT1 Gene in AML

WT1 (Figure 1) was initially identified as a tumor-suppressor gene involved in the pathogenesis of childhood renal Wilms’ tumor [6]. The gene is located on chromosome 11 (band 11p13) and encodes for a zinc finger DNA-binding protein with four major isoforms, each of which plays a significant role in normal gene function [5,7,8,9]. Physiologically, WT1 acts as a transcriptional factor, regulating the transcription of growth factors (such as PDGF-A chain, CSF-1, and IGF-II), growth factor receptors (IGF-IR), and other genes (such as RARA, c-myc, bcl-2) [5,10,11,12,13,14,15]. WT1 can either enhance or repress the expression of its target genes or constructs. In normal human bone marrow, WT1 expression is detected at extremely low levels and is restricted to the primitive CD34+ cell population [16]. In mouse models, it is thought to be involved in the self-renewal of early hematopoietic cells [5]. In normal human hematopoietic cells, WT1 appears to be a tumor-suppressor gene; indeed, its overexpression induces growth arrest, reduces colony formation, and promotes spontaneous differentiation [17,18].

Studies analyzing directly leukemic cells have shown that WT1 is highly expressed in the majority of AMLs, and even in blast crisis of chronic myeloid leukemia, whereas its expression is undetectable in normal blood cells [3,19,20]. Combining the results of several studies, WT1 RNA levels, as assessed by RT-PCR and Northern blot, were elevated in about 80% of AML patients [5,19,20,21,22,23,24]. In contrast to normal bone marrow (BM), in AML, WT1 overexpression appears to act as an oncogene and its reduction results in cell death [5]. Furthermore, its effect seems to be dependent on multiple protein partners such as p53, altering the pro-apoptotic behavior of both proteins, or growth factor signaling proteins such as FLT3, as AML with the FLT3-ITD mutation has been shown to be associated with the highest levels of WT1 [5,25,26]. It is currently unknown what drives WT1 overexpression in AML or if it is an early or late event in the disease onset. Moreover, it is unclear how a pro-apoptotic factor becomes an oncogene as it is unmutated, but it is likely due to a complex pattern of interactions [5]. Finally, the prognostic significance of WT1 overexpression is also matter of debate as several studies have yielded conflicting results [24,27,28,29].

## 3. The Crucial Role of MRD in AML

There is growing evidence that MRD detection is critical for assessing prognosis in AML, particularly in patients undergoing an intensive chemotherapy program [1]. The MRD monitoring is crucial to guide treatment after complete remission (CR), to define the need for consolidation with allogeneic stem cell transplantation (Allo-SCT), to detect impending relapse allowing early intervention, and to provide reliable post-transplant surveillance [1].

The MRD working group of the European LeukemiaNet (ELN) recently reviewed the main scientific evidence on MRD monitoring in AML and reached a consensus on thresholds and best timing to detect MRD, and also promoted the standardization of the different methods used [1,2].

To date, MRD can basically be measured using two methods, namely multiparametric flow cytometry (MFC) and real-time quantitative polymerase chain reaction (qPCR), for specific target genes or fusion transcripts: mutated NPM1, RUNX1-RUNX1T1, CBFB-MYH11, and PML-RARA. MRD surveillance based on next-generation sequencing (NGS) is an emerging and appealing technique but still under development [2,30].

MFC exploits the leukemia-associated immunophenotype (LAIP). It is applicable to more than 80% of patients, has a fast turnaround time, but is still operator-dependent. The approved cut-off is 0.1% but a MRD quantification below this threshold may be consistent with residual leukemia, and several studies have shown prognostic significance at lower cut-off levels of MFC-MRD [1,31,32]. The quantitative PCR for mutated NPM1, RUNX1-RUNX1T1, and CBFB-MYH11 is highly standardized and has a higher sensitivity than MFC (detection of one abnormal cell in 10^3^–10^6^ normal cells), but it can be used in only about 40% of AML patients. The most informative time points are after two cycles (in peripheral blood-PB samples) and at the end of treatment (in bone marrow-BM samples). 

In recent years, some studies explored a combination of MFC and WT1 for MRD monitoring in patients undergoing an intensive chemotherapy program, and these studies, albeit with different thresholds for WT1-MRD positivity, have shown that both markers have a high concordance and that the combination of MFC and WT1 can provide very important information to better stratify the prognosis of patients [33,34,35]. Because MFC is operator-dependent and the immunophenotypic shift in the leukemic clone may affect its power in predicting relapse, while the qPCR for WT1 is standardized and has limited operator bias, and mutations of the gene that can interfere with the analysis are rare, combining the two methods could add information to better stratify patients’ prognosis. Unfortunately, in the ELN consensus, WT1 expression plays a limited role as a MRD marker, and the authors recommended using this MRD marker only when no other MRD marker is present because of its low specificity and reduced sensitivity (about 10^−4^–10^−5^) [1]. However, based on our expertise with this MRD marker, we reassume the main experiences on MRD monitoring with WT1 overexpression, and we provide practical guidelines for the optimal dynamic use of this nonspecific molecular MRD marker in AML.

## 4. European LeukemiaNet Standardized Method for Quantitative Evaluation of WT1 Expression

In 2009, ELN researchers validated a quantitative WT1 assay and established reference ranges for WT1 expression in PB and BM analyzing a large number of control samples, to allow transcript levels indicative of residual leukemia to be distinguished from normal background levels [4]. The selected standardized assay (*Ipsogen WT1 ProfileQuant, QIAGEN*) is commercially available and includes exons 1 and 2, which are less prone to mutation than exons 7 and 9 (to reduce false-negative results). The upper normal values were set at 250 WT1 copies/10^4^ Abelson (ABL) for BM and at 50 WT1 copies/10^4^ ABL for PB, with a sensitivity of 10^−4^–10^−5^ [4].

## 5. Role of WT1 Expression Monitoring in Bone Marrow as MRD Marker

All studies reported in this section were performed in patients undergoing an intensive chemotherapy program (such as 3 + 7 or fludarabine-based regimens, followed by cytarabine-based consolidations, with or without allogeneic stem cell transplantation). WT1 overexpression was evaluated using the standardized method of Cilloni et al., but the proposed thresholds of normality in BM samples (less than 250 WT1 copies/10^4^ ABL) have not been widely accepted [4]. In Table 1, we summarize the selected studies together with the cut-off used.

With regard to pre-Allo-SCT setting, Cilloni et al. analyzed 91 patients with WT1 levels >20,000/10^4^ ABL at diagnosis and reported that the magnitude of WT1 log reduction after induction chemotherapy provided an independent predictor of relapse in the multivariate analysis, which remained highly significant even when patients were censored at the time of transplantation [4]. They also showed that detecting WT1 transcripts at levels above the upper limit of the normal value (WT1-MRD positive) at the end of consolidation predicted a significantly increased risk of relapse (67% vs. 42% at 5 years; *p* = 0.004).

The second is a study from our group, in which we analyzed 122 AML patients overexpressing WT1 at diagnosis and that underwent Allo-SCT in the first CR [36]. In this study, patients with WT1 levels within the normal range before Allo-SCT (WT1-MRD negative) had improved overall survival—OS (median not reached vs. 9 months, *p* < 0.0001) and disease-free survival—DFS (median not reached vs. 8 months, *p* < 0.0001) than those WT1-MRD-positive [36]. In addition, the relapse rate after Allo-SCT was 15% in patients WT1-MRD-negative and 44% in WT1-MRD-positive patients (*p* = 0.00073). WT1-MRD negativity was the only independent prognostic factor for improved OS and DFS in this study [36]. We confirmed the same prognostic relevance of WT1-MRD negativity before Allo-SCT in a subsequent study performed in FLT3-mutated AML, in which the median OS and DFS in the WT1-MRD-negative group were not reached and were 10.2 and 5.5 months, respectively, in the WT1-MRD-positive group (*p* = 0.0005 and *p* = 0.0001, respectively) [37]. It should be noted that, in this study, patients in CR who were WT1-MRD-positive had the same negative outcome as those without morphological CR [37].

In another study, Frairia et al. analyzed 255 AML patients who were overexpressing WT1 at diagnosis and who were tested for WT1 levels after induction and before Allo-SCT [38]. The authors reported that the median OS and DFS were significantly shorter in patients with >350 WT1 copies/10^4^ ABL after induction than in those with ≤350 WT1 copies (*p* = 0.018 and *p* = 0.025, respectively). Moreover, patients with WT1 > 150 copies before Allo-SCT had a significantly higher 2-year cumulative incidence of relapse (CIR) compared to those with WT1 ≤ 150 copies (HR 4.61, *p* = 0.002) [38].

Recently, Lambert et al. analyzed 341 AML patients treated in the ALFA-0702 trial, who were overexpressing WT1 at baseline and with available WT1 quantification after induction [39]. Both BM and PB were tested for WT1, and WT1-MRD positivity was defined when at least one of the two measurements was above the cut-off value (250 WT1 copies/10^4^ ABL for BM and at 50 WT1 copies/10^4^ ABL for PB). Post-induction WT1-MRD positivity after induction was predictive of subsequent relapse (4-year CIR 29% in WT1-MRD-negative vs. 61% in WT1-MRD-positive, *p* < 0.0001), and was an independent factor for CIR in the multivariate analysis [39]. In addition, at 4 years, relapse free survival—RFS was 60% in WT1-MRD-negative vs. 26% in WT1-MRD positive patients (*p* < 0.0001), and OS was 71% vs. 44% (*p* = 0.0005). In this study, WT1-MRD positivity remained independently associated with poorer RFS and OS in multivariate analysis, and the unfavorable prognostic significance of WT1-MRD positivity after induction was independent of Allo-SCT [39].

Finally, Nomdedéu et al. analyzed 584 patients in the CETLAM protocol (365 with available post-induction WT1 measurement and 287 with available post-consolidation WT1 measurement) and divided them into three groups according to post-induction WT1 levels (<17.5, 17.6 to 170.5, and >170.6/10^4^ ABL) and post-consolidation WT1 levels (<10, 10.1 to 100, and >100/10^4^ ABL) [40]. The median OS and DFS of the three post-induction groups were 59 and 59 months, 48 and 41 months, and 23 and 19 months, respectively. The median OS and DFS of the three post-consolidation groups were 72 and 65 months, 59 and 46 months, and 30 and 27 months, respectively. All differences between groups were statistically significant. Both post-induction and post-consolidation WT1 levels were significant for DFS and CIR in multivariate analysis [40].

WT1 has also been studied as an MRD marker in the post-Allo-SCT setting, mainly to identify and potentially treat early relapse. In a first study from our group, we analyzed 38 AML patients undergoing Allo-SCT with available quantitative WT1 evaluations before and after transplantation [41]. We observed a rapid decline in WT1 expression levels in all patients who achieved or maintained a CR after SCT. All patients who relapsed (13%) had increased WT1 expression at/or before relapse. We also found a complete concordance between WT1 expression levels and other MRD markers, when available [41]. In a subsequent study including 25 AML patients transplanted with reduced-intensity conditioning (RIC-Allo-SCT), we reported that cytological relapse was always anticipated by an increase in WT1 levels, and this increase anticipated the loss of molecular chimerism in 50% of the cases [42].

In another study, Pozzi et al. analyzed 122 AML patients with available WT1 evaluations before and after Allo-SCT, finding a higher relapse rate (54%) in patients with WT1 overexpression (exceeding 100 copies/10^4^ ABL) at any time post-SCT, as compared to patients with post-Allo-SCT WT1 expression <100 copies (16%, *p* < 0.0001) [43]. Similarly, the 5-year OS was 40% vs. 63%, respectively (*p* = 0.03). In multivariate analysis, WT1 overexpression post-Allo-SCT was the strongest predictor of relapse (HR 4.5, *p* = 0.0001) [43]. 

Using the same threshold of 100 copies of WT1/10^4^ ABL, Nomdedéu et al. reported that patients with <100 WT1 copies at the first evaluation after Allo-SCT had better outcomes in terms of OS, PFS, and CIR [44]. Additionally, in this study, patients with sustained WT1 levels under 100 copies showed a clear benefit in terms of OS, PFS, and CIR, even compared to patients with just a single measurement over 100 copies [44].

Finally, Duléry et al. used the standardized thresholds in BM and PB (250 copies/10^4^ ABL and 50 copies/10^4^ ABL, respectively) to evaluate 139 patients 3 months after Allo-SCT, and they found that WT1-MRD-positive patients at this time point had a poorer CIR (90% vs. 14.7%), EFS (at 3 years 10% vs. 72.3%), and OS (at 3 years 21.4% vs. 75.4%) than WT1-MRD-negative patients [45]. 

**Table 1 jcm-11-03306-t001:** Summary of studies exploring the role of WT1-MRD monitoring in BM.

*WT1-MRD Monitoring PRE-Allo-SCT)*			
Study (Reference)	WT1 Threshold	N° of Patients	Summary of Results
Cilloni et al. [4]	250 copies /10^4^ ABL	91	WT1 log reduction post-induction → independent predictor of relapse. WT1-MRD-positive post-consolidation → increased risk of relapse (67% vs. 42% at 5 years; *p* = 0.004).
Candoni et al. [36]	250 copies /10^4^ ABL	122	Better OS (median not reached vs. 9 months, *p* < 0.0001), DFS (median not reached vs. 8 months, *p* < 0.0001), and relapse rate (15% vs. 44%, *p* = 0.00073) for WT1-MRD-negative before Allo-SCT.
Candoni et al. [37]	250 copies /10^4^ ABL	62 FLT3 pos	Better OS (median not reached vs. 10.5 months, *p* = 0.0005) and DFS (median not reached vs. 5.5 months, *p* = 0.0001) for WT1-MRD-negative before Allo-SCT. Same outcome after Allo-SCT for WT1-MRD-positive and cases with active disease at the time of transplant.
Frairia et al. [38]	350 copies /10^4^ ABLpost induction 150 copies /10^4^ ABL pre-Allo-SCT	255	Shorter OS and DFS for WT1-MRD-positive after induction (HR for mortality 2.13, *p* = 0.018 and HR for relapse 2.81, *p* = 0.025). 2-year CIR after Allo-SCT higher for WT1-positive pre-Allo-SCT (HR 4.61, *p* = 0.002).
Lambert et al. [39]	250 copies /10^4^ ABL	341 (ALFA-0702 trial)	Better 4-year CIR (29% vs. 61%, *p* < 0.0001), RFS (60% vs. 26%, *p* < 0.0001), and OS (71% vs. 44%, *p* = 0.0005) for patients WT1-MRD-negative after induction.
Nomdedéu et al. [40]	3 groups post-induction (<17.5, 17.6 to 170.5, >170.6 /10^4^ ABL) and post-consolidation (<10, 10.1 to 100, >100 /10^4^ ABL)	365 (CETLAM protocol)	Median OS and RFS of the 3 post-induction groups: 59 and 59 months, 48 and 41 months, and 23 and 19 months, respectively. Median OS and RFS of the 3 post-consolidation groups: 72 and 65 months, 59 and 46 months, and 30 and 27 months, respectively.
** *WT1-MRD monitoring POST-Allo-SCT* **			
Candoni et al. [41]	250 copies /10^4^ ABL	38	Rapid decline in WT1 levels in patients in CR after SCT. WT1 increased before o at morphologic relapse in all patients.
Candoni et al. [42]	250 copies /10^4^ ABL	25	WT1 increased before relapse in all patients. WT1 increase anticipated loss of molecular chimerism in 50% of cases.
Pozzi et al. [43]	100 copies /10^4^ ABL	122	Higher incidence of relapse in WT1-MRD-positive at any time post-Allo-SCT. 5-year OS 40% for WT1-MRD-positive vs. 63% for WT1-MRD-negative cases (*p* = 0.03).
Nomdedéu et al. [44]	100 copies /10^4^ ABL	193	WT1-MRD-negative at first evaluation post-Allo-SCT had better OS, PFS, and CIR. Sustained WT1-MRD negativity → excellent outcomes.
Duléry et al. [45]	250 copies /10^4^ ABL	139	Worse CIR, EFS, and OS for WT1-MRD-positive than WT1-MRD-negative patients (90%, 10%, and 21.4% vs. 14.7%, 72.3%, and 75.4%, respectively). RIC associated with higher rate of WT1-MRD positivity at 3 months in univariate analysis (not in multivariate).

BM: bone marrow, OS: overall survival, DFS: disease-free survival, HR: hazard ratio, CIR: cumulative incidence of relapse, RFS: relapse-free survival, EFS: event-free survival, RIC: reduced intensity conditioning.

## 6. Role of WT1 Expression Monitoring in Peripheral Blood as MRD Marker

The main published studies exploring WT1 overexpression in PB as a marker for MRD in patients with AML are summarized in Table 2. Of note, in some of the studies cited in the previous section, WT1 expression was monitored in both BM and PB, and the results were reported together [4,39,45]. In all the reported studies, WT1 was analyzed in PB according to the standardized method of Cilloni et al. (cut-off value in PB samples: 50 WT1 copies/10^4^ ABL) [4]. 

In the study by Rautenberg et al., 64 AML/MDS patients were tested for WT1 levels in PB before Allo-SCT [46]. The 2-year post-transplant CIR, RFS, and OS were similar in WT1-MRD-positive and in refractory patients at transplant (61% vs. 70%, 37% vs. 26%, and 54% vs. 56%, respectively), but they were significantly better in WT1-MRD-negative cases (10%, 89%, and 90%, respectively) [46].

Another small retrospective study performed by Malagola et al., including 24 AML patients with available WT1 measurement in PB samples before Allo-SCT, found that patients with WT1 < 5 copies had a significantly better OS and DFS than patients with WT1 ≥ 5 copies (3-year OS 54% vs. 0%, *p* = 0.03; 1-year LFS 63% vs. 20%, *p* = 0.02) [47]. The relapse rate was 31% and 73%, respectively. In addition, the relapse rate was higher in patients with WT1 ≥ 5 copies at 3 and/or 6 months after Allo-SCT, and all patients with WT1 ≥ 5 copies pre-Allo-SCT, who did not have reduced levels at 3 or 6 months later, eventually relapsed [47].

In the post-Allo-SCT setting, Israyelyan et al. prospectively evaluated 82 patients with various myeloid malignancies (including 39 AML) and reported that, at a standard cut-off of 50 copies of WT1/10^4^ ABL, the method used for MRD monitoring had a specificity of 100% (positive predictive value 100%) and a sensitivity of 75% in detecting relapse [48]. Lower thresholds improved sensitivity, but the specificity decreased [48].

Lastly, Polak et al. reported complete concordance between the increase in WT1 expression and MRD positivity verified with alternative methods (MFC and chimerism or specific markers if available) in 32 AML patients after Allo-SCT. Notably, they also found that WT1 expression increases about 1 month earlier than the other markers [49].

**Table 2 jcm-11-03306-t002:** Summary of the studies exploring the role of WT1-MRD monitoring in PB.

*WT1-MRD Monitoring PRE-Allo-SCT)*			
Study (Reference)	WT1 Threshold	N° of Patients	Summary of Results
Cilloni et al. [4]	50 copies /10^4^ ABL	91 (15 only PB)	See Table 1. No differences between log reduction in PB compared to BM in patients with paired analysis available.
Lambert et al. [39]	50 copies /10^4^ ABL	341 (231 paired samples)	See Table 1. In paired samples, observed 9% of discrepancies. Discrepant patients were designed as WT1-MRD-positive.
Rautenberg et al. [46]	50 copies /10^4^ ABL	64 AML/MDS (50 AML)	Better 2-year CIR (10% vs. 61%, *p* < 0.01), RFS (89% vs. 37%, *p* < 0.01), and OS (90% vs. 54%, *p* = 0.03) for WT1-MRD-negative pre-Allo-SCT. Same outcome after Allo-SCT for WT1-MRD-positive and patients with active disease at the time of transplant.
Malagola et al. [47]	5 copies /10^4^ ABL	24	Better OS for WT1-MRD-negative (1-year: 81% vs. 60%, 2-year: 81% vs. 0%, 3-year: 54% vs. 0%; *p* = 0.03) and better RFS (1-year: 63% vs. 20%, *p* = 0.02). Relapse rate was 31% for WT1-MRD-negative patients pre-Allo-SCT vs. 73% for WT1-MRD-positive.
** *WT1-MRD monitoring POST-Allo-SCT* **			
Duléry et al. [45]	50 copies /10^4^ ABL	139 (106 PB)	See Table 1. Similar outcome for patients tested in PB and BM alone. Better correlation between relapse and WT1-MRD positivity in PB than in BM.
Malagola et al. [47]	5 copies /10^4^ ABL	24	Relapse rate higher in patients with WT1 ≥ 5 at 3 months (56% vs. 38%; *p* = 0.43) and 6 months (71% vs. 20%; *p* = 0.03) after Allo-SCT.
Israyelyan et al. [48]	50 copies /10^4^ ABL	82 (39 AML)	Specificity 100% with a positive predictive value of 100%, and sensitivity 75% for the method.
Polak et al. [49]	50 copies /10^4^ ABL	32	Absolute correlation with MFC, chimerism, and fusion transcripts. WT1 increase anticipated positivity by MFC and chimerism by 1 month.

BM: bone marrow, PB: peripheral blood, OS: overall survival, CIR: cumulative incidence of relapse, RFS: relapse-free survival, MFC: multiparameter flow cytometry.

## 7. WT1 Expression Assessment in Peripheral Blood Stem Cells Harvest

There are also a few studies analyzing WT1 expression in peripheral blood stem cell (PBSC) harvest, in patients eligible for autologous stem cell transplantation (Auto-SCT), to detect PBSC contamination with leukemic cells [50,51]. 

Messina et al. analyzed the PBSC of 30 AML patients who underwent Auto-SCT in the first CR [50]. The authors set a cut-off value of 80 WT1 copies/10^4^ ABL for PBSC, analyzing 22 PBSC harvests from controls (cases with multiple myeloma and lymphoproliferative disorders). According to this cut-off value, AML cases who received a WT1-positive PBSC graft (with >80 WT1 copies/10^4^ ABL) had a relapse incidence of 87% vs. 31% of patients who received a WT1-negative PBSC graft (*p* = 0.0001); OS and DFS were 25% and 25% for the WT1-positive group vs. 90% and 75% for the WT1-negative group (*p* = 0.0001) [50]. Interestingly, some patients who were WT1-MRD negative in BM before leukapheresis had a positive PBSC harvest, and most of these patients relapsed [50].

However, in another study including 72 AML patients, Mulé et al. did not observe differences in RFS between patients who received WT1-positive PBSC grafts and those who received WT1-negative PBSC grafts (using both the 80 and 50 WT1 copies/10^4^ ABL as cut-off values) [51]. The authors also reported that the administration of G-CSF could falsely enhance WT1 expression in PBSC even from healthy donors, a result in contrast to what was previously reported by Cilloni et al. and by Messina et al. [4,50,51]. 

## 8. Possible Pitfalls of WT1 Overexpression as MRD Marker

AML-MRD monitoring using WT1 overexpression, even if it is a nonspecific AML marker, can be easy to perform and very useful, but obviously, this marker also has limitations. 

It is recommended to refer to the standardized method of the ELN and its defined threshold (250 WT1 copies/10^4^ ABL on BM and 50 WT1 copies/10^4^ ABL on PB), which has been validated in large and multicenter cohorts of patients and normal controls, to have the best sensitivity and specificity to discriminate between normal and abnormal expression [1,4]. In addition, to use this MRD marker in the most appropriate way, it is always necessary to perform the evaluation of WT1 expression at AML diagnosis and to use the same sample (only BM or only PB) throughout the dynamic monitoring. The WT1 value at diagnosis gives us the sensitivity of the analysis, and the higher the value, the higher the sensitivity as the analysis can detect a higher logarithmic reduction after chemotherapy.

Even if the standardized ELN method (by Cilloni et al. [4]) targets WT1 exons 1 and 2, which are less prone to mutation than WT1 exons 7 and 9 (targeted by the previous tests), a very small number of false negative tests is possible due to mutations in exon 1 at the primer binding site [52]. This is a very rare event, but it must be considered. In the study by Cilloni et al., out of 504 patients analyzed, 32 had a WT1 expression below the cut-off value and only 3 of them had mutations at the primer binding site [4].

Another important aspect is that the reliability of the analysis also depends on the quality of the sample. As this is an RNA-based test, it is important that the RNA is undamaged prior to analysis and in adequate quantity to allow normal amplification of the gatekeeper gene ABL (which must reach 10^4^ copies). It is therefore evident that specific cases of AML, such as those occurring after hypoplastic myelodysplastic syndromes, or some transplanted patients, with low bone marrow cellularity or with bone marrow fibrosis, are more difficult to monitor with this method. 

## 9. Conclusions

We reviewed the key studies on WT1 overexpression as a marker for MRD in AML patients undergoing an intensive chemotherapy program, including Allo-SCT. Although most of these studies are retrospective and not easily comparable (due to differences in time-point used, treatment received by patients, and even thresholds for WT1-MRD positivity), WT1 may have important value and broad applicability, also taking into account the weakness that we previously mentioned. Nevertheless, this marker should be evaluated in prospective studies before it can be included in future guidelines for MRD monitoring in AML.

In conclusion, we can make some practical considerations about a proper use of WT1 as an MRD marker:WT1 is a non-specific panleukemic marker whose overexpression is detectable at diagnosis in about 80% of AML. WT1 can serve as a longitudinal MRD quantitative monitoring, alone (in AML without other biological markers) or in combination with other MRD markers (MFC and/or other specific markers if available).To evaluate the WT1 expression, it is recommended to use the standardized method of the European LeukemiaNet and its proposed cut-off values of 250 WT1 copies/10^4^ ABL on BM and 50 WT1 copies/10^4^ ABL on PB, which has been validated in large and multicenter cohorts of patients and normal controls (optimal sensitivity and specificity) [4].It is necessary to have the value of WT1 expression at diagnosis of AML, to exclude from the dynamic monitoring the cases of AML that do not overexpress WT1. WT1 monitoring should not be performed in subsequent treatment phases if WT1 expression is unknown at the time of AML diagnosis.To use WT1 as a MRD marker, we prefer the BM samples. However, monitoring WT1 expression from PB samples is feasible and simpler and may have accurate prognostic value, even considering the low reported discrepancy rate between the two sources [4,39].During treatment, the most important and informative time points for WT1-MRD evaluation are post-induction, to assess both the log reduction and achievement of WT1-MRD negativity, and before Allo-SCT, or at the end of consolidation, if Allo-SCT is not planned [4,36,37,38,39,40,46,47]. A recent large study (including 425 patients) conducted by Cho et al. supports this recommendation [53]. It is clear, from this study, that the most informative time point for WT1-MRD assessment was before Allo-SCT, and that the 250 copies/10^4^ ABL was the most predictive threshold for post-transplant relapse [53].In the post-transplant (or post-consolidation) setting, WT1-MRD evaluation can detect relapse earlier compared to other MRD methods (such as MFC and chimerism) and allow early intervention [42,49]. In this regard, some small studies, involving MRD-triggered ciclosporin withdrawal with or without donor lymphocyte infusions (DLI) or even hypomethylating agents, reported interesting results that need to be confirmed in prospective and larger clinical trials [43,54,55].There are still little data regarding WT1-MRD monitoring in PBSC grafts for patients that are candidates to Auto-SCT [50,51]. Further data are needed to better understand the role of WT1 monitoring in this setting.

## Figures and Tables

**Figure 1 jcm-11-03306-f001:**
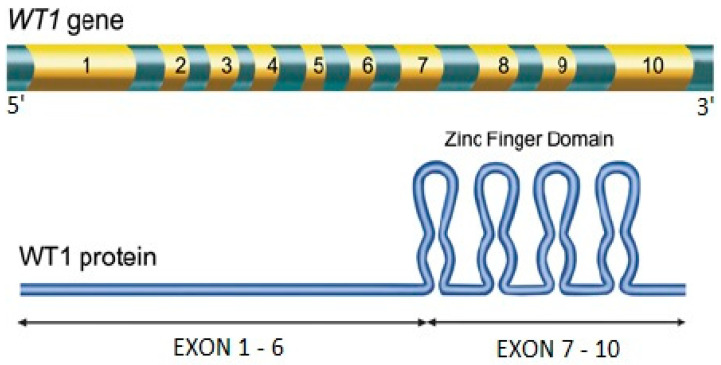
The WT1 gene.

## Data Availability

Not applicable.

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
