# Peer review of "The Role of Wilms’ Tumor Gene (WT1) Expression as a Marker of Minimal Residual Disease in Acute Myeloid Leukemia"

_jcm, 2022, doi:10.3390/jcm11123306_

Round 1

Reviewer 1 Report

e authors present a review article of the role of WT1 (Wilms tumor gene) as a marker for minimal residual disease in patients with acute myeloid leukemia. The topic is timely, and the paper is well written.

The study would be improved by some more detailed explanations such as the definition of intensive chemotherapy, whether myeloablative or reduced intensity transplant makes a difference.

1. Please explain why your data conflicts with the latest ELN recommendations.

2. Please define intensive chemotherapy.

3. What large groups, such as US BMT CTN, CIBMTR or EBMT, are collecting data on WT1 post transplant?

Author Response

The authors present a review article of the role of WT1 (Wilms tumor gene) as a marker for minimal residual disease in patients with acute myeloid leukemia. The topic is timely, and the paper is well written.

The study would be improved by some more detailed explanations such as the definition of intensive chemotherapy, whether myeloablative or reduced intensity transplant makes a difference.

Thank you for your questions. These are our answers:

  1. Please explain why your data conflicts with the latest ELN recommendations.

Our data does not conflict with ELN recommendations. The ELN recommendations take into account WT1 as MRD marker and all the reported studies in our paper followed the methodology for WT1 assessment supported by ELN. However, a small role is reserved for MRD monitoring with WT1 in ELN guidelines because this is not a specific MRD marker and, for this reason, some hematologic centers does not have experience with it. Currently, this marker needs to be evaluated in prospective studies for adult patients before it can be included in future guidelines for MRD monitoring in AML.  We clarified this aspect (lines 300-301)

  1. Please define intensive chemotherapy.

We added the definition on line 121 to 122. Regarding to the MAC vs RIC question, we reported in table 1 the only literature-available information.

  1. What large groups, such as US BMT CTN, CIBMTR or EBMT, are collecting data on WT1 post transplant?

We don't have this information. The CIBMTR has collected data on WT1 post-transplant, but in pediatric leukemias, that are not the subject of this review (ClinicalTrials.gov Identifier: NCT01385787).

Reviewer 2 Report

Dear authors,

thank you for this interesting overview. I have only one comment. The other, traditional MRD markers show expression in the AML clone itself, which WT1 expression does not or it is not clear. It is predictive as far as I understand you manuscript. Can you please integrate this minor point into your introduction/ discussion.

Kind regards

Author Response

Dear authors,

thank you for this interesting overview. I have only one comment. The other, traditional MRD markers show expression in the AML clone itself, which WT1 expression does not or it is not clear. It is predictive as far as I understand you manuscript. Can you please integrate this minor point into your introduction/ discussion.

Kind regards

Thank you for your comment. To clarify our paper, WT1 it is expressed at low levels in normal bone marrow and expressed at high levels in AML cells (as reported in lines 43 to 53). The level of expression makes the difference between normal and disease, as we reported in lines 110 to 117. In line 51-53 we clarified that the overexpression of WT1 is assessed on the leukemic cells.